# Phase-selective in-plane heteroepitaxial growth of H-phase CrSe$_2$

Meizhuang Liu [1,2] ✉, Jian Gou[2,3], Zizhao Liu[4], Zuxin Chen[5], Yuliang Ye[5], Jing Xu[5], Xiaozhi Xu [1], Dingyong Zhong [4], Goki Eda [2] & Andrew T. S. Wee [2] ✉

Phase engineering of two-dimensional transition metal dichalcogenides (2D-TMDs) offers opportunities for exploring unique phase-specific properties and achieving new desired functionalities. Here, we report a phase-selective in-plane heteroepitaxial method to grow semiconducting H-phase CrSe$_2$. The lattice-matched MoSe$_2$ nanoribbons are utilized as the in-plane heteroepitaxial template to seed the growth of H-phase CrSe$_2$ with the formation of MoSe$_2$-CrSe$_2$ heterostructures. Scanning tunneling microscopy and non-contact atomic force microscopy studies reveal the atomically sharp heterostructure interfaces and the characteristic defects of mirror twin boundaries emerging in the H-phase CrSe$_2$ monolayers. The type-I straddling band alignments with band bending at the heterostructure interfaces are directly visualized with atomic precision. The mirror twin boundaries in the H-phase CrSe$_2$ exhibit the Tomonaga-Luttinger liquid behavior in the confined one-dimensional electronic system. Our work provides a promising strategy for phase engineering of 2D TMDs, thereby promoting the property research and device applications of specific phases.

2D TMDs possess various polymorphic structures, including 2H (trigonal prismatic), 1 T (octahedral), 1 T' and T$_d$ phases, which are determined by the different chalcogen coordination geometries of metal atoms and stacking orders of atomic layers[1]. The polymorphic phases give rise to a variety of intriguing properties such as superconductivity[2,3], ferroelectricity[4], ferromagnetism[5,6], Weyl semimetal[7], and quantum spin Hall effect[8]. The physical and chemical properties of TMDs can be modulated by changing the structural phases. The so-called phase engineering of TMDs makes it possible to achieve the precise control of structural phases in a desired manner[9–11]. Phase engineering can be implemented through two main strategies of phase transformation and phase-selective synthesis. The phase transformation strategies, including alkali metal intercalation[12,13], alloying[14], thermal treatment[15,16], electric field[17], mechanical strain[18,19], and external irradiation[20,21] have been employed to trigger the structural transformation to effectively

obtain the target phases. The aforementioned chemical and physical methods might encounter the issue of impurities and defects introduced during the phase transformation process as well as the metastable phases recovering to their original structures when the external stimuli are removed[22]. In comparison, phase-selective synthesis has the ability to directly fabricate 2D TMDs with a specific phase and high phase purity. The strategies of precursor design, atmosphere regulation, and temperature control have been utilized in chemical vapor deposition (CVD) growth to achieve phase-selective synthesis[23–26]. As for the molecular beam epitaxy (MBE) growth, it is highly desirable to develop general and controllable strategies to fabricate 2D TMDs with excellent phase selectivity.

Heteroepitaxial templates have been employed to seed the growth of crystals with unconventional phases[27–30]. The heteroepitaxial template usually provides a constrained environment to guide the

[1]School of Physics, Guangdong Basic Research Center of Excellence for Structure and Fundamental Interactions of Matter, Guangdong Provincial Key Laboratory of Quantum Engineering and Quantum Materials, South China Normal University, Guangzhou 510006, China. [2]Department of Physics, National University of Singapore, 2 Science Drive 3, 117542 Singapore, Singapore. [3]School of Physics, Zhejiang University, Hangzhou 310027, China. [4]School of Physics and State Key Laboratory of Optoelectronic Materials and Technologies, Sun Yat-sen University, Guangzhou 510275, China. [5]School of Semiconductor Science and Technology, South China Normal University, Guangzhou 510631, China. ✉ e-mail: mzliu@m.scnu.edu.cn; phyweets@nus.edu.sg

crystal growth with the formation of lattice-matched interfaces. Under the effect of heteroepitaxial templates, a stabilized nonequilibrium phase can be formed within a critical coherent thickness, for instance, the metastable cubic phase of aluminum nitride (AlN) was stabilized in AlN/TiN superlattice films[27,31]. The formation of a lattice-matched interface with lower interfacial energy is the essential prerequisite for the stable existence of metastable phases. The heteroepitaxial templates play a prominent role in phase control of noble metal nanocrystals, for example, the unconventional 4H-phase Ag, Pd, Pt, Ir, Rh, Os, Ru, and Cu were synthesized using 4H-Au nanoribbons as the epitaxial templates[30,32]. In two-dimensional TMD materials, due to the existence of atomically flat dangling-bond-free surfaces, in-plane hetereoepitaxial template effects are considered to seed the growth of 2D TMDs for achieving the selective growth of metastable phase structures in MBE growth.

In this work, we achieved the phase-selective MBE growth of H-phase $CrSe_2$ induced by the in-plane template of $MoSe_2$ nanoribbons. The in-plane heteroepitaxial template plays a crucial role in the phase-selective growth of H-phase $CrSe_2$ by minimizing the interfacial energy by lattice-matching. Distinct from the magnetic metallic T-phase $CrSe_2$, the nonmagnetic semiconducting character of the H-phase $CrSe_2$ is revealed in our work. The atomically sharp interfaces in the $MoSe_2$-$CrSe_2$ lateral heterostructures and the characteristic defects of mirror twin boundaries in the H-phase $CrSe_2$ are directly observed by scanning tunneling microscopy (STM) and non-contact atomic force microscopy (nc-AFM). The band alignments and band bending at the $MoSe_2$-$CrSe_2$ lateral and $CrSe_2$/$MoSe_2$ vertical heterostructure interfaces are detected using the scanning tunneling spectroscopy. The mirror twin boundaries (MTBs) in the H-phase $CrSe_2$ exhibit the quantum-confined Tomonaga–Luttinger liquid behavior, which includes charge density modulation, length-dependent bandgap opening, and spin-charge separation.

## Results

### Phase-selective growth of H-phase $CrSe_2$

Atomically thin $CrSe_2$ films are usually observed in the metallic 1T phase, which is demonstrated by atomic-resolution transmission electron microscopy and other experiments[6,33–35]. As shown in the schematic illustration (Fig. 1a), the codeposition of Cr and Se atoms on a substrate of highly oriented pyrolytic graphite (HOPG) will produce the metallic T-phase $CrSe_2$ islands. The T-phase $CrSe_2$ has an octahedral coordination structure with $D_{3d}$ symmetry and degenerate $d_{xy,yz,xz}$ and $d_{x^2-y^2,z^2}$ orbitals. The partially filled $d$ orbitals give rise to the metallic electronic properties as illustrated in Fig. 1b. By utilizing the in-plane heteroepitaxial template of H-phase $MoSe_2$ nanoribbons, the growth of H-phase $CrSe_2$ monolayers can be guided with minimizing the interfacial energy (Fig. 1a). The atomically matched lattices guarantee the formation of lateral heterostructures with seamlessly connected interfaces. Unlike metallic T-phase $CrSe_2$, the H-phase $CrSe_2$ adopts the trigonal prismatic coordination structure with $D_{3h}$ symmetry, which splits the $d$ orbitals into three groups of $d_{z^2}$, $d_{x^2-y^2,yz}$ and $d_{xz,yz}$ with a sizeable bandgap (Fig. 1c).

In the STM image of the MBE-grown $CrSe_2$ samples (Fig. 1d), the T-phase $CrSe_2$ islands exhibit approximately hexagonal shapes with straight edges. A nonlayered growth behavior is unraveled by the existence of a CrSe thin layer with a height of ~0.3 nm shown in Fig. 1d. The lattice constant is calibrated to be $3.4 \pm 0.1$ Å, which is consistent with that of previously reported 1T-$CrSe_2$ crystals (3.39 Å) synthesized through deintercalation of lithium from $LiCrSe_2$[35]. The metallic characteristic of the T-phase $CrSe_2$ monolayer is unveiled in the differential conductance ($dI/dV$) spectrum (Fig. 1g). In order to achieve the phase-selective growth of 1H $CrSe_2$, the in-plane epitaxial template of $MoSe_2$ nanoribbons are prepared beforehand with the typical width of 10–50 nm at a growth temperature of about 550 °C (Fig. 1e). The distinctive Mo- and Se-edges at the opposite sides of nanoribbons can be distinguished by different edge features in the atom-resolved STM images (Supplementary Fig. 1). After the growth of $MoSe_2$ nanoribbons, Cr and Se atoms are subsequently codeposited onto the same HOPG substrate with the substrate temperature kept at about 200 °C. The diffusing atoms preferentially nucleate and aggregate at the active edges of $MoSe_2$ nanoribbons due to the existence of dangling bonds at the edges. The different growth temperatures and deposition duration will lead to the different surface morphologies (Supplementary Fig. 2). The H-phase $CrSe_2$ rather than T-phase structure is formed with the lattice matching under the effect of the in-plane epitaxial template of $MoSe_2$ nanoribbons. The as-grown H-phase $CrSe_2$ segments are seamlessly fused to $MoSe_2$ nanoribbons with the formation of lattice-matched lateral heterostructures (Fig. 1f). In the lateral heterostructures; the continuous H-phase interface structures have lower interfacial energy compared with the 1H–1T interface structures as revealed by density functional theory (DFT) calculations (Supplementary Fig. 3). The chemical states of Cr and Se elements in two distinct phases of $CrSe_2$ were investigated by X-ray photoelectron spectroscopy (XPS). The measured binding energies of Cr $2p$ and Se $3d$ electrons (Fig. 1h) show an energy shift of -0.5 eV between H- and T-phase $CrSe_2$ due to the different electronic structures. The electronic properties of H-phase $CrSe_2$ are revealed by STS measurements, which exhibit a semiconducting character with a bandgap of $0.75 \pm 0.05$ eV. The increased peak intensity of local electronic states with the invariant bandgap can be observed in the $dI/dV$ spectra taken at different tunneling currents (Fig. 1i). The semiconducting property of H-phase $CrSe_2$ monolayer is also verified by DFT calculations. The electronic structure of H-phase $CrSe_2$ manifests a direct bandgap of 0.72 eV with the valence band maximum and conduction band minimum derived mostly from the Cr $3d$ orbitals (Supplementary Fig. 3). The nonmagnetic property of H-phase $CrSe_2$ has been revealed in the previously reported works by DFT calculations[36,37]. In our X-ray magnetic circular dichroism (XMCD) measurements, no obvious ferromagnetic signals were detected for the H-phase $CrSe_2$ in both normal incidence (NI) and grazing incidence (GI) directions at 78 K (Supplementary Fig. 4).

### Structural characterization of $MoSe_2$–$CrSe_2$ interfaces

The atomically sharp interfaces of $MoSe_2$–$CrSe_2$ lateral heterostructures are revealed both by STM and nc-AFM. In the STM image of the $MoSe_2$–$CrSe_2$ lateral heterostructure (Fig. 2a), $CrSe_2$ segments are seamlessly connected to the $MoSe_2$ nanoribbons in between. The different STM image contrast between $CrSe_2$ and $MoSe_2$ primarily originates from the local electronic states rather than topographic features, which varies with the applied bias voltages (Supplementary Fig. 5). There is a continuous linear defect of mirror twin boundary (MTB) crossing through the heterostructure interface, reflecting the in-plane heteroepitaxial template effect. The occurrence of MTB characteristic defects in $CrSe_2$ provides strong evidence for the claim of H-phase $CrSe_2$. The atomic structures and electronic properties of MTBs in $CrSe_2$ will be further elaborated in more detail later. The corresponding ball-and-stick model of the lateral heterostructure with an MTB line defect is exhibited in Fig. 2b. In the atomic row of MTB, fourfold rings share a point at the chalcogen site leading to each Se atom bound to four Cr atoms instead of three. The Mo- or Se-edge of the $MoSe_2$ nanoribbon can be determined by the orientation of MTB due to the constraint of mirror symmetry. To detect the strain at the heterostructure interfaces, the moiré pattern in the $MoSe_2$ region was studied since it can be used as a magnifying glass to directly visualize the lattice-misfit strain at the heterostructure interfaces[38]. In the high-resolution STM image (Fig. 2c), the $MoSe_2$ region exhibits an undistorted $3 \times 3$ moiré pattern, indicating that there is no obvious lattice-misfit strain at the heterostructure interface. The nc-AFM technique was also employed to characterize the interface structures as it can provide surface topography without being disturbed by electronic

states. The constant-height nc-AFM frequency shift images (Fig. 2d, e) taken at Se- and Mo-edges of MoSe₂ nanoribbons, respectively (labeled in Fig. 2a), both exhibit atomically sharp interface structures without any dislocations. The nc-AFM observations also revealed the atomically sharp MoSe₂–CrSe₂ interface structures at the adjacent Mo-edge

and Se-edge with the included angle of 120° (Supplementary Fig. 6). These results imply that CrSe₂ in the lateral heterostructure adopts the same H-phase structure as MoSe₂. If we postulate that it was T-phase CrSe₂ connected to the different edges of MoSe₂ nanoribbons, dislocations and lattice misfit strain would emerge at the interfaces[21] due

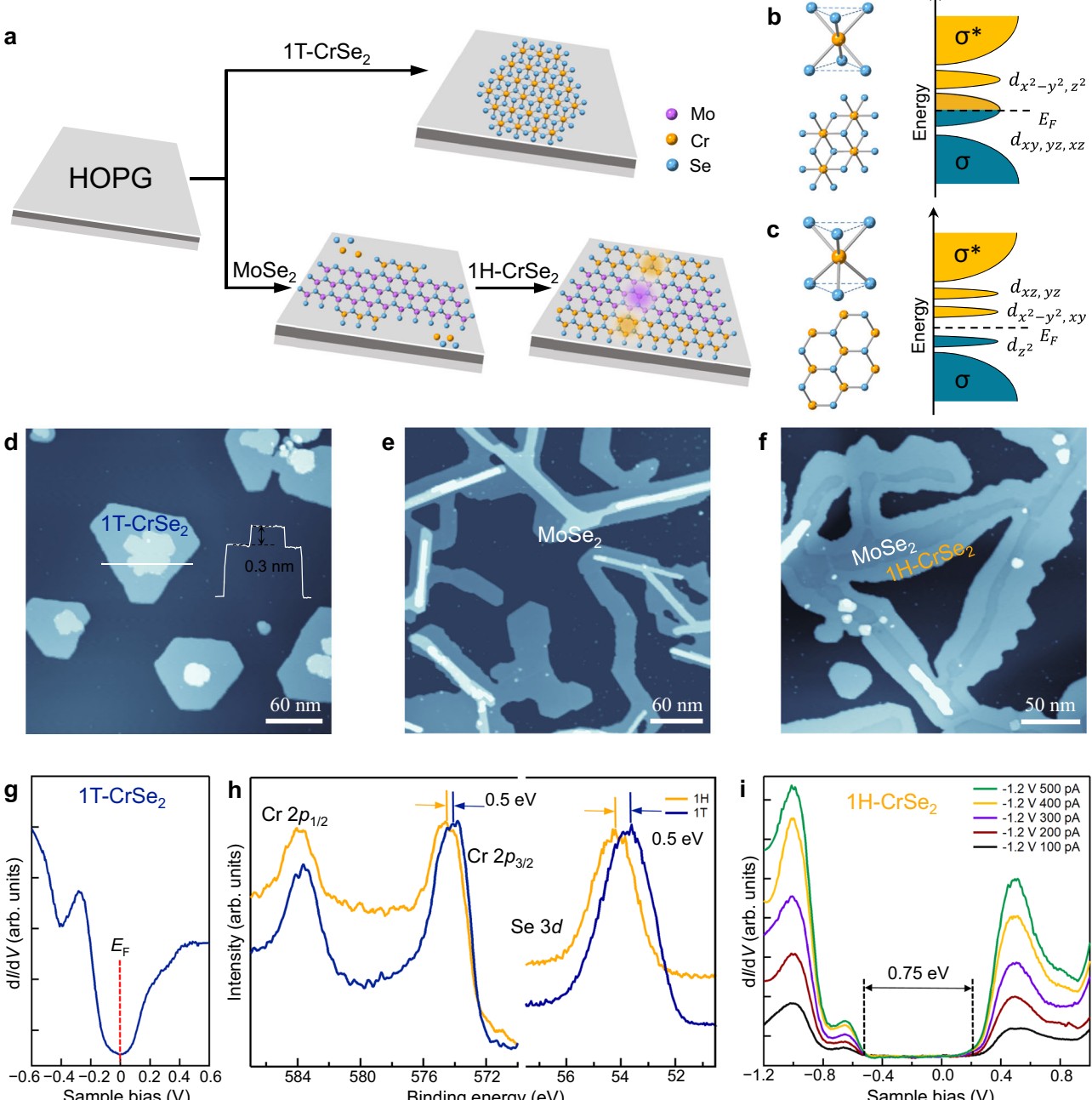

**Fig. 1 | Phase-selective growth of H-phase CrSe₂ monolayers induced by the in-plane template of MoSe₂ nanoribbons. a** Schematic illustration of the epitaxial growth of T-phase and H-phase CrSe₂ at different growth conditions. The highly oriented pyrolytic graphite (HOPG) is chosen as the epitaxial substrate. In the atomic model of the lateral heterostructure, the purple area represents the MoSe₂ nanoribbon, and the yellow areas represent the CrSe₂ segments. **b,c** The ball-and-stick models and electron energy diagrams for the T-phase CrSe₂ and H-phase CrSe₂, respectively. In the electron energy diagrams, $d_{x^2-y^2,z^2}$, $d_{xy,yz,xz}$, $d_{xz,yz}$, $d_{x^2-y^2,yz}$ and $d_{z^2}$ represent the different $d$ orbitals that are located within the bandgap between the bonding (σ) and antibonding (σ*) states, and $E_F$ indicates the Fermi level. **d** Scanning tunneling microscopy (STM) topographic image of an isolated 1T-CrSe₂ island on a HOPG substrate (sample voltage $V_S = -2.0$ V, tunneling

current $I_t = 10$ pA). The inset shows the height profile across the CrSe₂ island. **e** STM topography of MoSe₂ nanoribbons grown at 550 °C ($V_S = -2.0$ V, $I_t = 10$ pA). **f** Large-scale STM image of lateral heterostructures with H-phase CrSe₂ segments seamlessly connnected to MoSe₂ nanoribbons ($V_S = 1.3$ V, $I_t = 10$ pA). **g** Differential conductance (d$I$/d$V$) spectrum measured on T-phase CrSe₂. The vertical dashed line at the 0 V sample voltage indicates the Fermi level. **h** X-ray photoelectron spectroscopy (XPS) characterization of the Cr 2$p$ and Se 3$d$ peaks in T- and H-phase CrSe₂ monolayers. **i** Differential conductance (d$I$/d$V$) spectra taken at the same bias voltage and different tunneling currents on the CrSe₂ regions of lateral heterostructures. The bandgap is marked by the vertical dashed lines at the position of the valence band maximum and conduction band minimum, respectively.

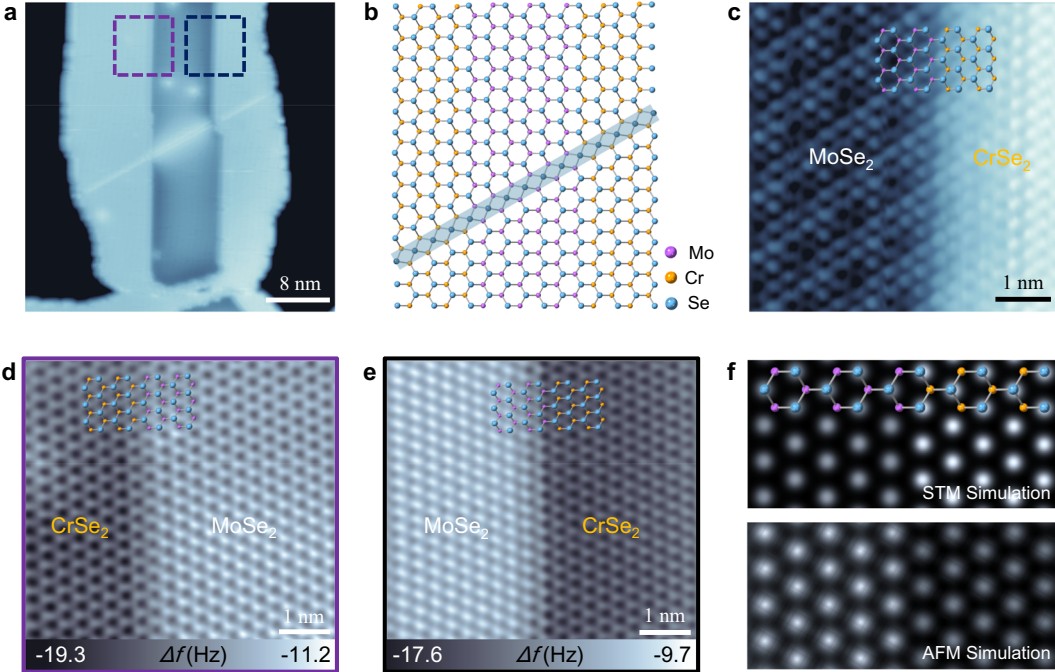

**Fig. 2 | Structural characterization of the MoSe₂–CrSe₂ lateral hetero-structures. a** STM image of the MoSe₂–CrSe₂ lateral heterostructure with a continuous mirror twin boundaries (MTB) linear defect crossing through the interface ($V_S = -1.0$ V, $I_t = 3$ pA). **b** The corresponding ball-and-stick model of the lateral heterostructure with an MTB line defect. **c** High-resolution STM image of the atomically sharp interface with $3 \times 3$ moiré pattern in the MoSe₂ region ($V_S = -0.4$ V, $I_t = 0.6$ nA). **d, e** Constant-height non-contact atomic force microscopy (nc-AFM) images of the MoSe₂–CrSe₂ interfaces taken at the Se-edge and Mo-edge of MoSe₂ nanoribbon with partly overlaid structure models, respectively. The scanning areas are labeled by the purple and black dashed squares in (**a**), respectively (Tip height $z = -360$ pm relative to the height at the setpoint 1.3 V, 10 pA). **f** Density functional theory (DFT) simulated STM and constant-height nc-AFM images with the corresponding structure model.

to the different lattice symmetries and lattice constants as observed in the 1H–1T CrSe₂ interfaces (Supplementary Fig. 7). In the nc-AFM images, the MoSe₂ regions have a brighter image contrast (higher frequency shift) compared with CrSe₂ regions, which can be ascribed to the different Pauli repulsive forces at small tip-sample distances. We adopted the model of MoSe₂–CrSe₂ lateral heterostructure with continuous H-phase structures at the interfaces to simulate the STM and nc-AFM images. The observed STM and nc-AFM image contrast between the MoSe₂ and CrSe₂ regions are reproduced well by the DFT simulation (Fig. 2f).

## Band alignment in the lateral and vertical heterostructures

STS measurements were carried out to explore the spatial evolution of electronic structures of both lateral and vertical heterostructures at the atomic scale, which is of fundamental importance for understanding the interface properties and evaluating their application potentials. The close-up STM image (Fig. 3a) displays the MoSe₂–CrSe₂ lateral heterostructures with 12 nm-width MoSe₂ nanoribbon in between. The type-I band alignment with a straddling gap is found for the MoSe₂–CrSe₂ lateral heterostructures, which endows the lateral heterostructures with the potential for light-emitting applications and studying light-matter interaction. The band profile across the interface is directly visualized by the color rendering of the d$I$/d$V$ mapping (Fig. 3b). A bandgap of ~2.30 eV with Fermi level located 1.60 eV above the valence band maximum (n-type) is detected in the MoSe₂ region away from the interface. The MoSe₂ region exhibits upward band bending for both valence and conduction bands near the interface due to the charge transfer from MoSe₂ to CrSe₂. The band profile at the interface shows a sharp transition into the CrSe₂ region, which has a smaller bandgap of ~0.75 eV. A slight downward band bending can be seen in the magnified band profile of the CrSe₂ region (Fig. 3c), which is attributed to electron injection from MoSe₂. The valence and conduction band offset (VBO

and CBO) in the MoSe₂–CrSe₂ lateral heterostructures are determined to be -0.84 eV and -0.78 eV, respectively. Besides the band bending, the effect of bandgap narrowing emerges at the heterostructure interface, probably due to strain at the interface. As illustrated in Fig. 3d, the downward shift of the conduction band is larger than that of the valence band, resulting in a bandgap narrowing of ~0.06 eV for CrSe₂. The inconsistent upward band shifts in valence and conduction bands of MoSe₂ also lead to a bandgap narrowing of ~0.10 eV.

The formation of CrSe₂/MoSe₂ vertical heterostructures is based on the bilayer MoSe₂ nanoribbons. In the bilayer MoSe₂ nanoribbons, the second-layer (SL) MoSe₂ nanoribbons usually have a narrower width than the first-layer (FL) nanoribbons. The epitaxial growth of H-phase CrSe₂ will take place simultaneously on both layers, with the SL-CrSe₂ seamlessly connected to the in-between SL-MoSe₂ nanoribbons (Fig. 3e). The representative CrSe₂/MoSe₂ vertical heterostructures are formed with the second layer being the H-phase CrSe₂ and first layer being the MoSe₂ in the bilayer structures. The spatially resolved d$I$/d$V$ spectra are acquired along the path with a step size of 8.5 Å labeled in Fig. 3e. The band offsets and edge states at the interface can be intuitively observed in the energy profile of the CrSe₂/MoSe₂ vertical heterostructure (Fig. 3f). The valence and conduction bands of the first-layer MoSe₂ exhibits a joint upward band bending at the interface with a magnitude of ~0.14 eV and ~0.07 eV, respectively. Compared with the small band bending in the previously reported bilayer-monolayer MoSe₂ and WSe₂ homostructures[39], the band shifts in the first-layer MoSe₂ indicate a stronger interlayer interaction in the vertical CrSe₂/MoSe₂ heterostructures. Regarding the second-layer CrSe₂ region, the valence and conduction bands are bent upward by ~0.28 and ~0.36 eV, respectively (magnified in Supplementary Fig. 8). The edge state with a narrow gap (marked by orange lines in Fig. 3f) emerges at the edge termination of second-layer CrSe₂ due to the existence of dangling bonds. The STS mapping is further sliced into

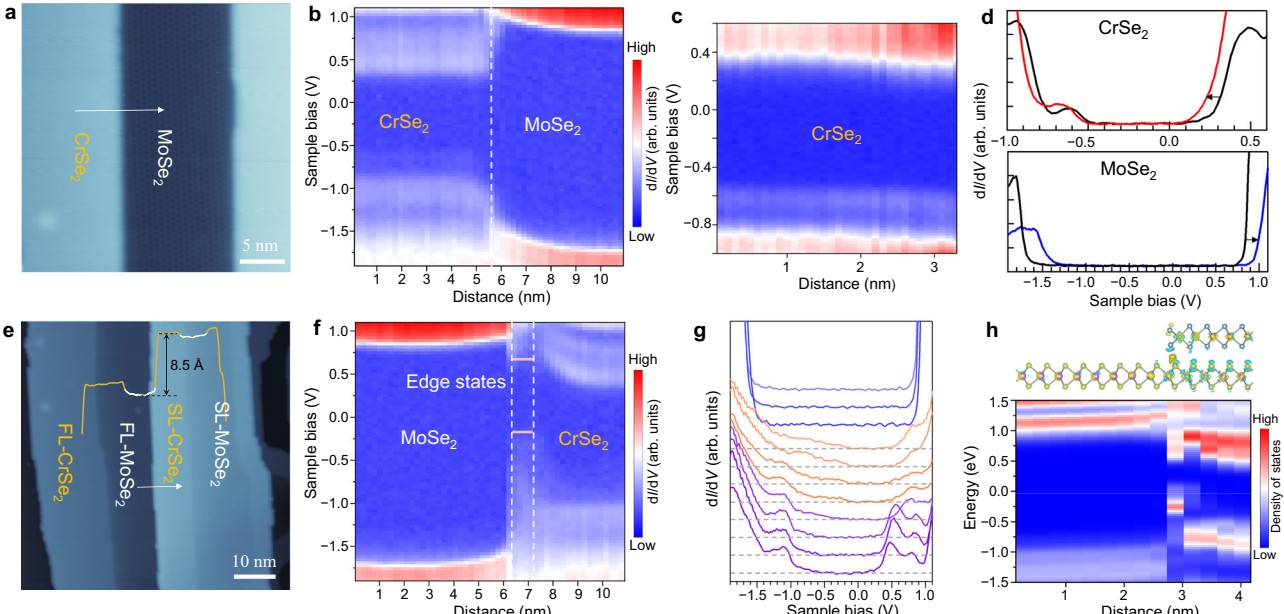

**Fig. 3 | Electronic structures at the interfaces of MoSe₂–CrSe₂ lateral and CrSe₂/MoSe₂ vertical heterostructures. a** STM topography of MoSe₂–CrSe₂ lateral heterostructures with straight interfaces ($V_S = -1.3$ V, $I_t = 30$ pA). **b** Real-space imaging of the band profile of the MoSe₂–CrSe₂ lateral heterostructure plotted in terms of Log (d$I$/d$V$). The d$I$/d$V$ spectra were acquired along the white arrow crossing the interface with a total length of 11 nm. The vertical dashed line indicates the interface of the lateral heterostructure. **c** The magnified band profile in the CrSe₂ region exhibits a slight downward band bending near the interface. **d** The d$I$/d$V$ spectra taken on the CrSe₂ and MoSe₂ regions near the interface (red line for CrSe₂, blue line for MoSe₂) compared with the ones away from the interface (black lines). The arrows indicate the band shifts in the d$I$/d$V$ spectra. **e** Large-scale STM image of the

CrSe₂/MoSe₂ vertical heterostructure formed with the second-layer (SL) CrSe₂ and first-layer (FL) MoSe₂ in the vertical structure ($V_S = -1.3$ V, $I_t = 30$ pA). The inset shows the height profile across the heterostructure. **f** 2D plot of the d$I$/d$V$ spectra across the step of the CrSe₂/MoSe₂ vertical heterostructure. The d$I$/d$V$ spectra were taken from −1.9 V to 1.1 V along the white arrow in (**e**) with a total length of about 11 nm. The edge state with a narrow gap is marked by the vertical dashed lines and orange lines. **g** Selected subset of the d$I$/d$V$ spectra taken along the white arrow. The d$I$/d$V$ spectra are offset for clarity. **h** DFT-calculated band structures at the interface of the vertical heterostructure. The calculated differential charge density with the side view of the corresponding structure model is shown in the upper panel.

selected individual spectra obtained at the vicinity of the interface (Fig. 3g). In the d$I$/d$V$ spectra of second-layer CrSe₂, the original characteristic peak around −0.6 eV turns to be a sloping bump with energy position ~0.18 eV lower than that of monolayer CrSe₂ on the semimetallic HOPG substrate. The increase of bandgap in the second-layer CrSe₂ can be attributed to the better electronic isolation provided by the underlying MoSe₂ interlayer. At the heterostructure interface, the localized dangling-bond states give rise to the new spectral features with smaller bandgaps in the d$I$/d$V$ curves (as shown in Fig. 3g). The edge states and band shifts in the vertical heterostructure are further verified by our DFT-calculated local density of states (LDOS) plot of the Se atoms on the surface (Fig. 3h). The calculated differential charge density reveals the charge accumulation at the edge termination which induces the upward band bending of second-layer CrSe₂ near the edge. The upward band bending in the first-layer MoSe₂ due to the interlayer charge transfer can also be reproduced by DFT calculations (Supplementary Fig. 8).

## Electronic properties of MTBs in the H-phase CrSe₂

MTBs, as quasi-one-dimensional metals, provide an ideal platform for exploring the electronic behavior in the confined system. MTBs with a 4|4 P structure have been extensively studied in H-phase molybdenum dichalcogenides such as MoSe₂ and MoTe₂[40–43] but rarely detected in other non-Mo-based TMD materials. Isolated MTBs with triangular shape (Fig. 4a) are obtained in the H-phase CrSe₂ monolayers. The close-up high-resolution STM image of MTB (Fig. 4b) exhibits a typical feature of two bright parallel lines, as previously observed in MoSe₂ and MoTe₂, but without an obvious spatial modulation period. The structure model of the MTB is shown in Fig. 4c, and the corresponding atomic structure is revealed by the nc-AFM image (Fig. 4d), in which a

surface feature that a row of Se atoms have a darker contrast than other Se atoms in the region can be observed. The darker contrast mainly results from the higher density of Cr atoms in the MTB, which gives rise to stronger attractive forces. The d$I$/d$V$ spectra (Fig. 4e) taken on the MTB with a length of ~12.8 nm exhibit a feature of gap opening at the Fermi level. The gap size increases with the length of MTBs getting shorter, which is one of the signatures of Tomonaga-Luttinger liquid (TLL) behavior (Supplementary Fig. 9). In the TLL theory, the energy gap of the finite system with length $L$ can be described as $E_{gap} = [(\pi v_c/2K_c) + (\pi v_s/2K_s)](1/L)$, where $v_c$ and $v_s$ stand for the velocity of charge and spin excitation, respectively. Two Luttinger parameters $K_c$ and $K_s$ encode the interaction strength[42,44].

The STS spectra acquired on the MTB exhibit the metallic characteristic with low-energy states around the Fermi level (Fig. 4e). In the magnified d$I$/d$V$ spectrum (inset in Fig. 4e), four pronounced peaks located at 37.6, 15, −12, and −34.2 mV can be identified. The constant-height d$I$/d$V$ conductance mappings (Fig. 4g) measured at the four different peak energies reveal a periodic charge density modulation at the period of ~3a (triple lattice constant of CrSe₂). The modulation period is also observed in the low-bias constant-height STM image (Fig. 4f), which reflects the spatial distribution of low-energy electronic states. As shown in Fig. 4g, the spatial modulation of the highest occupied state (HOS) and lowest unoccupied state (LUS) is in phase with a symmetric two-lobe feature in the MTB, but it is partially out of phase for the second occupied and unoccupied states. The observed charge density modulation may not be a consequence of charge density waves (CDW) but the quantum confinement of electrons in the finite-length MTBs. In the d$I$/d$V$ spectra along the MTB (Fig. 4h), the periodic fluctuation of peak positions at the negative bias from −13.2 mV to −5.7 mV and at the positive bias from 11.3 mV to 15.4 mV

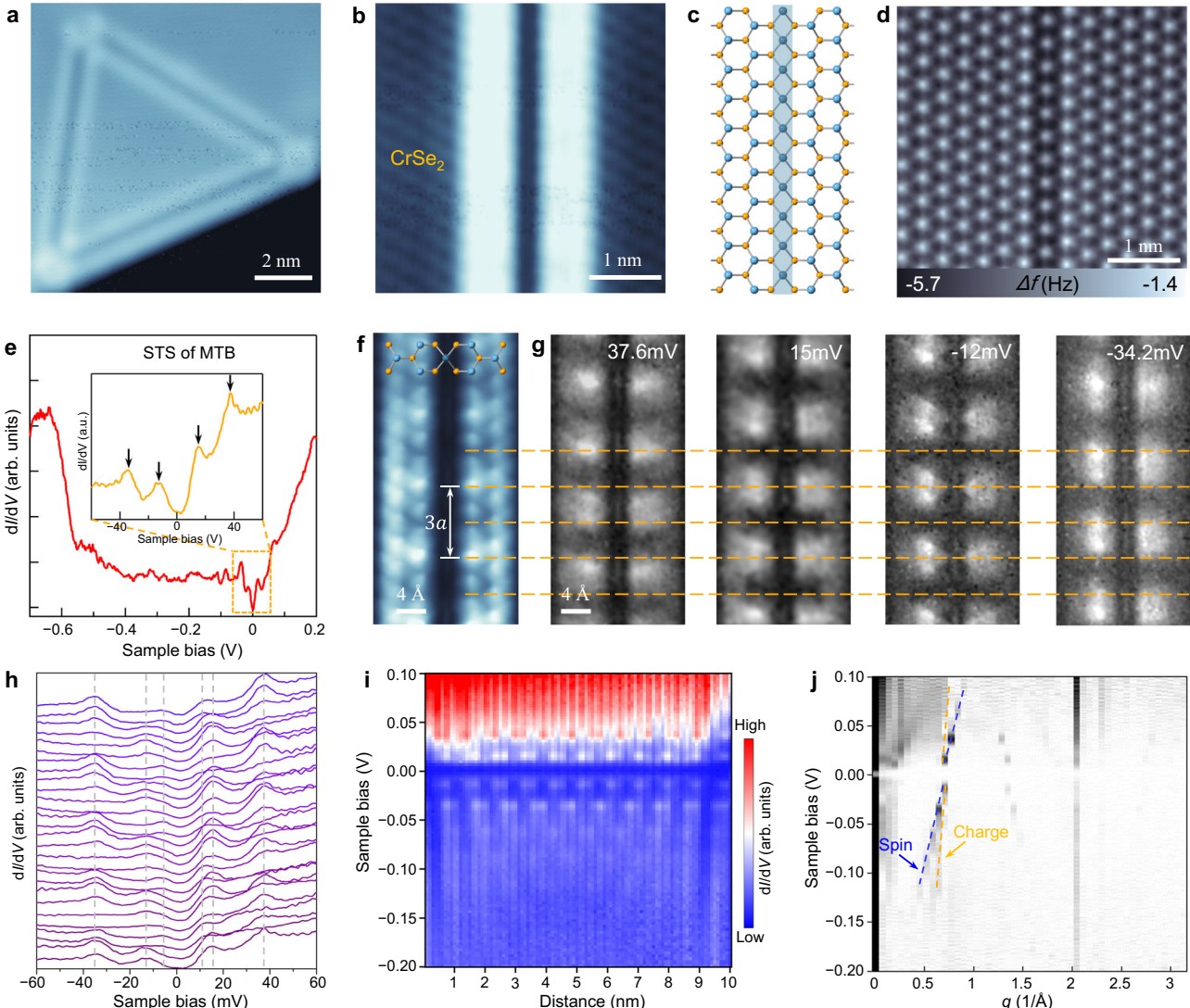

**Fig. 4 | MTBs in the H-phase CrSe$_2$ monolayers. a** STM image of a triangular MTB loop in the CrSe$_2$ monolayer ($V_S = -1.3$ V, $I_t = 10$ pA). **b** Close-up STM image of the MTB appearing as bright double rows ($V_S = 100$ mV, $I_t = 10$ pA). **c** Structure model of the MTB with four-fold rings sharing a point at the chalcogen site. **d** Atomic-resolution nc-AFM image of an MTB in CrSe$_2$ (Tip height $z = -80$ pm relative to the height at the setpoint 100 mV, 10 pA). **e** The d$I$/d$V$ spectrum taken on the MTB with a length of -12.8 nm from $-0.6$ V to 0.2 V. The inset shows the magnified d$I$/d$V$ spectrum from $-60$ mV to 60 mV. **f** Low-bias constant-height STM image of the MTB with an intensity modulation at the period of -3a ($V_S = 18$ mV, Tip height $z =$ $-80$ pm relative to the height at the setpoint 100 mV, 10 pA). **g** Constant-height STS mappings measured at the four energies of 37.6 mV, 15 mV, $-12$ mV and $-34.2$ mV, respectively. The horizontal dashed lines indicate the position corresponding to that in (**f**). **h** d$I$/d$V$ spectra taken along the bright line of the MTB. **i** Real-space imaging of band profile of the MTB. **j** Fourier transform of the d$I$/d$V$ data in (**i**) as a function of sample bias V and wavevector $q$. Two linear dispersion branches with different velocities are marked by blue and yellow dashed lines, which corresponds to the spin-charge separation predicted by Tomonaga–Luttinger liquid theory.

for the HOS and LUS respectively can be discerned. The discrete quantized states below HOS and above LUS are directly visualized in the color plot of d$I$/d$V$ spectra along the MTB (Fig. 4i). The phenomena may be described by TLL theory that low-energy excitations are standing waves with discrete energies in a finite-length MTB system[44]. Moreover, the TLL behavior of spin-charge separation is observed in the Fourier transform of the 2D STS map (Fig. 4j), in which two linear dispersion branches corresponding to the spin and charge density excitations have different slopes (marked with blue and yellow lines)[42].

## Discussion

In this work, we successfully achieved the in-plane template-induced selective growth of H-phase CrSe$_2$ with the formation of MoSe$_2$–CrSe$_2$ lateral and MoSe$_2$/CrSe$_2$ vertical heterostructures. The atomically sharp interfaces in the MoSe$_2$–CrSe$_2$ lateral heterostructures and the

characteristic defect of MTBs in the CrSe$_2$ monolayer are unveiled by the atomic-resolution STM and nc-AFM, verifying the H-phase structure in CrSe$_2$. The same phase structure and lattice constant of the H-phase CrSe$_2$ and MoSe$_2$ ensure that no obvious lattice-misfit strain exists at the interfaces of MoSe$_2$–CrSe$_2$ lateral heterostructures. The H-phase CrSe$_2$ has the semiconducting character with a bandgap of $0.75 \pm 0.05$ eV for the monolayer on the HOPG substrate and an increase of bandgap to a magnitude of -0.93 eV for the second-layer CrSe$_2$ on the MoSe$_2$ interlayer. Visualization of band profiles of the lateral and vertical heterostructures allows the identification of band alignment and band bending at the interfaces. In addition, the MTBs in H-phase CrSe$_2$ exhibit the quantum confined TLL behavior, including the charge density modulation, length-dependent band-gap opening, and spin-charge separation. Our study paves the way for phase-selective growth of 2D-TMDs by in-plane heteroepitaxial templates, enabling further property research and device applications on 2D TMDs.

## Methods

### Experimental measurement

The MoSe$_2$–CrSe$_2$ heterostructures were fabricated by MBE growth in an ultrahigh vacuum chamber (base vacuum $8 \times 10^{-10}$ mbar). The HOPG substrate was cleaved and then transferred into an ultrahigh vacuum chamber to degas at 600 °C. Molybdenum and chromium atoms were evaporated from the electron-beam evaporators. The excessive selenium atoms were sublimed at the temperature of 160 °C to maintain the Se-rich environment. MoSe$_2$ nanoribbons are formed by codeposition of Mo and Se atoms with the substrate temperature kept at 550 °C. The heterostructures can be obtained through the subsequent growth of CrSe$_2$ at the substrate temperature of about 200 °C. The lateral growth rate of H-phase CrSe$_2$ is about 0.02 ML/min in our experiments. The substrate temperature from 180 to 250 °C is proper for the growth of 1H-CrSe$_2$. The STM and nc-AFM measurements were performed on an Omicron low-temperature STM system operated at 78 K. The STM images were acquired in the constant-current mode, and the nc-AFM measurement was carried out in constant-height frequency modulation mode with a native tungsten tip without CO functionalization. The AFM simulation is based on the online modeling software provided by Hapala et al.[45] Differential conductance (d$I$/d$V$) spectra of MTBs were acquired at 4.3 K measured by a lock-in detection with a 963 Hz and 1–5 mV modulation superimposed on the sample bias. WSxM software was used to process all STM and AFM images[46].

### Theoretical calculation

Spin-polarized density functional theory calculations were performed using the projector augmented wave method for valence-core interactions, Perdew–Burke–Ernzerhof (PBE) functional of generalized gradient approximation, and a plane-wave basis set as implemented in the Vienna ab-initio Simulation Package (VASP)[47–49]. The plane-wave kinetic energy cutoff was set to 500 eV for geometric and electronic structure calculation. All the atoms were allowed to fully relax until the force on each atom was less than 0.01 eV $\tilde{A}^{-1}$. The vacuum region was set to 20 Å to avoid spurious interactions between periodic images. The Hubbard U was taken into account to describe the on-site Coulomb repulsion of Cr's 3$d$ electrons.

## Data availability

Relevant data supporting the key findings of this study are available within the article and the Supplementary Information file. All raw data generated during the current study are available from the corresponding authors upon request.

## Code availability

The computer code used for data processing is available from the corresponding authors upon request.

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

## Acknowledgements

This work was supported by the Guangdong Basic and Applied Basic Research Foundation (grant No. 2022A1515111001 (M.L.)), the Science and Technology Program of Guangzhou (grant No. 2024A04J0002 (M.L.)) and the National Research Foundation of Singapore (MOE Tier 2 grant WBS A-8000942-00-00 (A.W.)). The computation part of the work was supported by the National Supercomputer Center in Guangzhou.

## Author contributions

M.L. and A.T.S.W. proposed and conceived this project. M.L. and J.G. performed the STM and nc-AFM experiments. M.L., Z.L., and J.X. conducted the MBE growth. M.L. and Y.Y. provided theoretical support for the experiments. M.L., Z.C., X.X, D.Z., G.E., and A.T.S.W. did the data analysis and discussed the results. M.L., D.Z., and A.T.S.W. wrote the paper with the comments from all co-authors.

## Competing interests

The authors declare no competing interests.
