## [Peer Review File · Nature Communications]

Phase-selective in-plane heteroepitaxial growth of H-phase
 CrSe_2REVIEWER COMMENTS

Reviewer #1 (Remarks to the Author):

In this paper, the authors realized the growth of 1H-CrSe₂ (as opposed to the common 1T-CrSe₂) through in-plane templating with 1H-MoSe₂. The authors performed a systematic study combining STM and nc-AFM to confirm the structural phase of 1H-CrSe₂ and studied the electronic structure at the 1H-CrSe₂/1H-MoSe₂ interface. I read the paper with joy. The experimental data is of high quality, and the result is convincing. The paper is very well written, and the overall idea of an in-plane template for MBE growth is very smart, which could open possibilities for phase engineering other 2D materials with epitaxial growth not limited to MBE, but also other methods. I support the publication of this paper in Nature Communication.

I have a few questions regarding the paper and would like to discuss them with the authors. I believe these questions could further improve the quality of the paper.

1. How critical is the growth temperature to realize 1H-CrTe₂? Will a mixed phase form if the growth temperature is too low/ too high?
2. Did the author observe transition from 1H-CrSe₂ to 1T-CrSe₂ when the 1H islands are significantly large? Did the authors observe 1T-1H CrSe₂ boundaries?
3. Did the authors observe isolated CrSe₂ grown on top of MoSe₂? What is the phase?
4. Do the authors expect any magnetism in 1H-CrSe₂?
5. There is a typo in Fig. 3e, where the step height should be 8.5Å instead of 8.5 nm.

Reviewer #2 (Remarks to the Author):

This manuscript reported the in-plane template induced selective synthesis of H-phase CrSe₂ in the molecular beam epitaxy (MBE) growth. The authors used the lattice-matched MoSe₂ nanoribbons as the in-plane heteroepitaxial template to seed the growth of H-phase CrSe₂ with minimizing the interfacial energy by lattice-matching. The authors present the high-resolution scanning tunneling microscopy and non-contact atomic force microscopy images, which reveal the atomically abrupt interfaces in the MoSe₂-CrSe₂ lateral heterostructures and the characteristic defect of MTBs in the H-phase CrSe₂ monolayers. The STS measurements directly visualized the band alignments of the lateral and vertical heterostructures in atomic precision with obvious band bending at the interfaces. The quantum confined Tomonaga-Luttinger liquid behavior was revealed in the characteristic defect of MTBs in H-phase CrSe₂. MTBs in non-Mo-based TMD materials will provide the platform for exploring the intriguing properties of confined one-dimensional electronic system. The ideas proposed by this manuscript are novel and interesting. Some issues need to be addressed before publication:

- 1) Some reported strategies for phase engineering are only effective for the specific materials. How is the universality of this method for phase- selective growth of 2D TMDs?
- 2) The magnetism of chromium selenide has recently attracted much attention. How is the magnetic

property of the H-phase CrSe₂?

3) Fig. 2c: please overlay the atomic model on the STM image for easier identification.

4) Fig. 3d: the x-axis should represent the sample bias not distance.

5) Is the band bending in the first-layer MoSe₂ of CrSe₂/MoSe₂ vertical heterostructure due to the existence of edge states or the charge transfer between two layers?

Reviewer #3 (Remarks to the Author):

This study experimentally demonstrated in-plane MoSe₂ template induced selective growth of H-phase semiconducting CrSe₂ and the decisive role of in-plane heteroepitaxial template with interfacial energy by lattice matching. The visualization of band profiles of the in-plane heterostructures between MoSe₂ and CrSe₂ combined with theoretical calculations seem attractive because a universal integration scheme for 2D/2D lateral heterostructures can guarantee the possibilities of design and fabrication of various (opto-)electronic 2D devices. In general, this is a rigorous and well-organized work and the findings are of considerable interest to scientists working in the 2D materials/devices field. However, this reviewer noticed that the concept of in-plane template induced selective growth has already been demonstrated in previous studies and well developed. Thus, this reviewer suggests the authors to convincingly illustrate the novelty of the current work. At least, the following issues should be carefully addressed:

i) First of all, it is better to provide the underlying mechanism related to basic nucleation and growth mechanism of 1D MoSe₂ nanoribbons at the reported growth conditions (i.e., Why can the MoSe₂ crystals be successfully obtained in the form of 1D nanoribbons?)

ii) The authors are also asked to provide the detailed information on the structural properties (width, thickness, etc.) of the 1D MoSe₂ templates depending on the growth parameters (at their optimized growth regime) owing to their great important role in the lateral heteroepitaxy and please further explain what the main variables were that were able to control the thickness of the single-crystalline 1D MoSe₂ at the monolayer level.

iii) In Fig. 1f, why are the edges of as-grown H-CrSe₂ crystals strongly jagged and not atomically straight during MBE growth? (and what was the lateral growth rate of H-CrSe₂?) In addition, why does CrSe₂ prefer edge-mediated growth compared to surface-mediated growth? Not even the presence of CrSe₂ island above the MoSe₂? The reasons must be explained in detail.

iv) Page 7, line 189: Why does CrSe₂/MoSe₂ vertical heterostructure form and under what growth conditions? In this case, I believe the second-layer CrSe₂ is H-phase based on the supporting STS study and then how H-CrSe₂ crystals can be grown on MoSe₂ instead of the growth T-CrSe₂? And the authors claim it follows a layer-by-layer growth mode as compared to the lateral growth of H-CrSe₂. Though, no details on growth conditions and underlying mechanism are provided.

i) The quantum-confined TLL behavior depending on the length of MTBs, quasi-1D metals, in H-phase CrSe₂ needs to be explained in detail, which will provide direction toward further research and device applications.

We thank the reviewers for the positive evaluation and constructive suggestions of our work. Their perspective has definitely helped us to further improve the quality of our manuscript. We provide a point-by-point response to the reviewers' questions and comments. The corresponding changes made to the manuscript are indicated by underlines in blue.

Point-by-point response to the reviewers' comments:

Reviewer #1 (Remarks to the Author):

In this paper, the authors realized the growth of 1H-CrSe₂ (as opposed to the common 1T-CrSe₂) through in-plane templating with 1H-MoSe₂. The authors performed a systematic study combining STM and nc-AFM to confirm the structural phase of 1H-CrSe₂ and studied the electronic structure at the 1H-CrSe₂/ 1H-MoSe₂ interface. I read the paper with joy. The experimental data is of high quality, and the result is convincing. The paper is very well written, and the overall idea of an in-plane template for MBE growth is very smart, which could open possibilities for phase engineering other 2D materials with epitaxial growth not limited to MBE, but also other methods. I support the publication of this paper in Nature Communication.

I have a few questions regarding the paper and would like to discuss them with the authors. I believe these questions could further improve the quality of the paper.

1. How critical is the growth temperature to realize 1H-CrSe₂? Will a mixed phase form if the growth temperature is too low/ too high?

Reply: We thank the review for the positive evaluation and constructive suggestions on our work. The range of growth temperature from 180-250°C is proper for realizing the 1H-CrSe₂. In our experiment, excessive selenium atoms were sublimed to maintain the Se-rich environment during the growth. The lowest growth temperature was set at 120 °C for thermally desorbing the excessive Se atoms from the substrate. At the growth temperature of 120 °C, only low-quality film of chromium selenide can be obtained (as shown in Fig. R1a). Thermal energy is essential for atoms diffusing and nucleating to form the better films. However, when the growth temperature is higher than 300 °C, most samples form Cr₂Se₃, whose heights are 0.3 nm higher than that of MoSe₂ (Fig. R1b). Meanwhile, a small portion of CrSe₂ was observed to have well-ordered 1D patterns (Fig. R1c and 1d). This can be ascribed to the annealing-induced Se-deficient linear defects, which is similar with the spontaneous formation of well-ordered 1D patterns in the VSe₂ monolayers by annealing at high temperature of 350 °C (*Nano Lett.* 2019, 19, 4897–4903). In our opinion, the control of growth temperature is very critical for realizing 1H-CrSe₂ structures due to the existence of diverse structures at the different growth temperatures.

Fig. R1 | Samples of chromium selenide grown at 120°C and 350°C. **a**, STM image of chromium selenide grown at 120 °C ($V_s = -2$ V, $I_t = 10$ pA). **b**, STM image of MoSe₂ together with Cr₂Se₃ islands grown at 350 °C ($V_s = -2.2$ V, $I_t = 8$ pA). **c**, STM image of CrSe₂ on the first-layer MoSe₂ with the well-ordered 1D patterns ($V_s = -2.0$ V, $I_t = 50$ pA). **d**, Atomic-resolution STM image of annealing-induced 1D patterns in CrSe₂ ($V_s = 0.2$ V, $I_t = 1.3$ nA).

Change1. Added two panels (Fig. R1a and 1b) in Supplementary Fig. 2 exhibiting the samples of chromium selenide grown at 120°C and 350°C. The Supplementary Note “STM characterization of subsequently grown chromium selenide at different growth temperature and deposition duration” was added.

Change2 (Page 10). Added the sentence “The substrate temperature from 180-250°C is proper for the growth of 1H-CrSe₂.” into Methods.

2. Did the author observe transition from 1H-CrSe₂ to 1T-CrSe₂ when the 1H islands are significantly large? Did the authors observe 1T-1H CrSe₂ boundaries?

Reply: We did observe the 1H-1T CrSe₂ boundaries. As shown in Fig. R2a and 2b, the 1H and 1T phases can be distinguished by the different surface morphologies due to the different electronic properties. As a result of the different phase structures and lattice constants between 1H and 1T phases, the rough interfaces with misfit dislocations can be observed in Fig. R2c and 2d. The formation of 1H-1T CrSe₂ boundaries might result from two mechanisms: first, the in-plane template effects cannot occur beyond the critical distance and, second, the formed 1T-CrSe₂ islands diffuse and merge with the 1H-CrSe₂ islands. But it’s hard for us to judge which mechanism is preferred. In our experiments, the size of 1H-CrSe₂ islands will be limited because of the existence of pre-prepared MoSe₂ nanoribbons on the surface.

Fig. R2 | STM images of 1H-1T CrSe₂ boundaries. **a**, STM image of the MoSe₂-CrSe₂ lateral heterostructure with 1H-1T CrSe₂ boundaries ($V_S = 2$ V, $I_t = 10$ pA). **b**, STM image of the heterostructure with 1H-1T CrSe₂ boundaries ($V_S = -1.2$ V, $I_t = 80$ pA). **c**, Close-up STM image of the 1H-1T CrSe₂ interface ($V_S = -0.6$ V, $I_t = 300$ pA). **d**, Atomic-resolution STM image of the 1H-1T CrSe₂ interface ($V_S = -0.3$ V, $I_t = 600$ pA). **e**, Atomic-resolution STM image of the 1H-1T CrSe₂ interface ($V_S = -0.1$ V, $I_t = 600$ pA). **f**, Height profile taken along the red and black lines in **e**.

Change3. Added Supplementary Fig. 7 with figure captions exhibiting the STM images of 1H-1T CrSe₂ boundaries.

Change4 (Page 6). Replaced the sentence “however, they are not observed in our experiments” with the sentence “as observed in the 1H-1T CrSe₂ interfaces (Supplementary Fig. 7)”.

3. Did the authors observe isolated CrSe₂ grown on top of MoSe₂? What is the phase?

Reply: In our experiments, the isolated CrSe₂ islands grown on the top of MoSe₂ can be observed by increasing the deposition duration of CrSe₂ (as shown in Fig. R3). Most samples are the 1T phase with non-layered structures.

Fig. R3 | STM images of CrSe₂ grown on the top of MoSe₂. a,b, STM images of isolated 1T-CrSe₂ islands grown on the top of MoSe₂ ($V_s = -1.8$ V, $I_t = 30$ pA).

Change5. Added two panels (Fig. R3a and 3b) in Supplementary Fig. 2 exhibiting the samples of isolated 1T-CrSe₂ islands grown on the top of MoSe₂.

Change6 (Page 4). Added the sentence “The different growth temperatures and deposition duration will lead to the different surface morphologies (Supplementary Fig. 2)” in the paragraph.

4. Do the authors expect any magnetism in 1H-CrSe₂?

Reply: The nonmagnetic semiconducting property of H-phase CrSe₂ has been revealed in the previously reported works by DFT calculations (*J. Phys. Chem. C* 119, 13169–13183, 2015; *Cryst. Res. Technol.* 58, 2200274, 2023). The X-ray magnetic circular dichroism (XMCD) measurements were carried out to detect the magnetic properties of H-phase CrSe₂ and MoSe₂ in the lateral heterostructures. The XMCD signal, $\Delta\mu = \mu_+ - \mu_-$, is obtained from the difference between X-ray absorption spectra (XAS) at different helicities of circularly polarized light, μ_+ and μ_- . No obvious ferromagnetic signals were detected for the H-phase CrSe₂ in both normal incidence (NI) and grazing incidence (GI) directions at 78 K.

Fig. R4 | XAS/XMCD spectra of the H-phase CrSe₂ and MoSe₂ at 78 K. XAS/XMCD spectra of H-phase CrSe₂ measured in the normal incidence (NI) a and grazing incidence (GI) b directions at 78 K. c, XAS/XMCD spectra of H-phase MoSe₂ in the MoSe₂-CrSe₂ lateral heterostructures measured at 78 K.

Change7. Added **Supplementary Fig. 4** with the three panels (Fig. R4a, b and c) exhibiting the XMCD spectra of the H-phase CrSe₂ and MoSe₂ at 78 K.

Change8 (Page 5). Added the sentence “The nonmagnetic property of H-phase CrSe₂ has been revealed in the previously reported works by DFT calculations^{36,37}. And in our X-ray magnetic circular dichroism (XMCD) measurements, no obvious ferromagnetic signals were detected for the H-phase CrSe₂ in both normal incidence (NI) and grazing incidence (GI) directions at 78 K.” in the paragraph.

5. There is a typo in Fig. 3e, where the step height should be 8.5Å instead of 8.5 nm.

Reply: We thank the review for the careful reading and kind reminding. We revised the unit of the step height and checked the article carefully.

Change9. The unit of the step height in Fig. 3e was revised to be Å instead of nm.

Reviewer #2 (Remarks to the Author):

This manuscript reported the in-plane template induced selective synthesis of H-phase CrSe₂ in the molecular beam epitaxy (MBE) growth. The authors used the lattice-matched MoSe₂ nanoribbons as the in-plane heteroepitaxial template to seed the growth of H-phase CrSe₂ with minimizing the interfacial energy by lattice-matching. The authors present the high-resolution scanning tunneling microscopy and non-contact atomic force microscopy images, which reveal the atomically abrupt interfaces in the MoSe₂-CrSe₂ lateral heterostructures and the characteristic defect of MTBs in the H-phase CrSe₂ monolayers. The STS measurements directly visualized the band alignments of the lateral and vertical heterostructures in atomic precision with obvious band bending at the interfaces. The quantum confined Tomonaga-Luttinger liquid behavior was revealed in the characteristic defect of MTBs in H-phase CrSe₂. MTBs in non-Mo-based TMD materials will provide the platform for exploring the intriguing properties of confined one-dimensional electronic system. The ideas proposed by this manuscript are novel and interesting. Some issues need to be addressed before publication:

1) Some reported strategies for phase engineering are only effective for the specific materials. How is the universality of this method for phase-selective growth of 2D TMDs?

Reply: We thank the review for the positive evaluation and constructive suggestions. To verify the universality of the in-plane template induced phase-selective growth, the lattice-matched VSe₂ is selected to grow together with MoSe₂ nanoribbons to construct the lateral heterostructures. In the MoSe₂-VSe₂ lateral heterostructure, VSe₂ segments are seamlessly connected to the in-between MoSe₂ nanoribbon (Fig. R5d and 5e). The MTBs that are the characteristic defects of H-phase TMDs also emerge in the VSe₂ regions labelled by white square frames, indicating the formation of H-phase VSe₂. The atomic structure of MTBs in the H-phase VSe₂ monolayer can be directly resolved by STM imaging (Fig. R5f). In comparison, it is almost impossible to obtain the atomic structures of MTBs in the MoSe₂ samples through STM imaging due to the large discrepancy in the local density of electronic states. The H-phase VSe₂ monolayer has a well-order hexagonal lattice, which is distinct from the ($\sqrt{3} \times 2$) and ($\sqrt{3} \times \sqrt{7}$) charge-density-wave superstructures in the monolayer 1T-VSe₂ confirmed by our STM and nc-AFM observations (Fig. R5b and c). The distinct electronic and magnetic properties of

H-phase VSe₂ still need further experimental exploration. We believe this in-plane template induced phase-selective growth can be a promising and universal approach for phase engineering of 2D TMDs, thereby further expanding the library of crystal phases and promoting the property research of specific phases.

Fig. R5 | Selective growth of H-phase VSe₂ monolayers by in-plane heteroepitaxial template effects. **a**, STM image of monolayer 1T-VSe₂ with a strong stripe modulation ($V_s = -0.03$ V, $I_t = 1$ nA). **b**, Atomic-resolution STM of 1T-VSe₂ with $(\sqrt{3} \times 2)$ and $(\sqrt{3} \times \sqrt{7})$ CDW superstructures ($V_s = 0.01$ V, $I_t = 1.5$ nA). **c**, Constant-height nc-AFM image of 1T-VSe₂. **d,e**, Seamlessly fused MoSe₂-VSe₂ lateral heterostructures with the emergence of MTBs in VSe₂ regions. **f**, Atomic-resolution STM image of MTBs in the VSe₂ region ($V_s = 20$ mV, $I_t = 1.8$ nA).

2) The magnetism of chromium selenide has recently attracted much attention. How is the magnetic property of the H-phase CrSe₂?

Reply: The same question on the magnetic property of H-phase CrSe₂ was also raised by the review #1. The nonmagnetic semiconducting property of H-phase CrSe₂ has been revealed in the previously reported works by DFT calculations (*J. Phys. Chem. C* 119, 13169–13183, 2015; *Cryst. Res. Technol.* 58, 2200274, 2023). The X-ray magnetic circular dichroism (XMCD) measurements were carried out to detect the magnetic properties of H-phase CrSe₂ and MoSe₂ in the lateral heterostructures. The XMCD signal, $\Delta\mu = \mu_+ - \mu_-$, is obtained from the difference between X-ray absorption spectra (XAS) at different helicities of circularly polarized light, μ_+ and μ_- . No obvious ferromagnetic signals were detected for the H-phase CrSe₂ in both normal incidence (NI) and grazing incidence (GI) directions at 78 K.

Fig. R4 | XAS/XMCD spectra of the H-phase CrSe₂ and MoSe₂ at 78 K. XAS/XMCD spectra of H-phase CrSe₂ measured in the normal incidence (NI) **a** and grazing incidence (GI) **b** directions at 78 K. **c**, XAS/XMCD spectra of H-phase MoSe₂ in the MoSe₂-CrSe₂ lateral heterostructures measured at 78 K.

Change7. Added **Supplementary Fig. 4** with the three panels (Fig. R4a, b and c) exhibiting the XMCD spectra of the H-phase CrSe₂ and MoSe₂ at 78 K.

Change8 (Page 5). Added the sentence “The nonmagnetic property of H-phase CrSe₂ has been revealed in the previously reported works by DFT calculations^{36,37}. And in our X-ray magnetic circular dichroism (XMCD) measurements, no obvious ferromagnetic signals were detected for the H-phase CrSe₂ in both normal incidence (NI) and grazing incidence (GI) directions at 78 K.” in the paragraph.

3) Fig. 2c: please overlay the atomic model on the STM image for easier identification.

Reply: The atomic model of MoSe₂-CrSe₂ heterostructure was overlaid for easier identification.

Change10. The atomic model of MoSe₂-CrSe₂ heterostructure was overlaid onto Fig. 2c.

4) Fig. 3d: the x-axis should represent the sample bias not distance.

Reply: We thank the review for the careful reading and kind reminding. We revised the label of x-axis in Fig. 3d to sample bias.

Change11. The label of x-axis in Fig. 3d was changed to sample bias.

5) Is the band bending in the first-layer MoSe₂ of CrSe₂/MoSe₂ vertical heterostructure due to the existence of edge states or the charge transfer between two layers?

Reply: To verify that the band bending is due to the charge transfer instead of edge states, DFT calculations were carried out. The atomic model (Fig. R6a) with the finite-

width CrSe₂ nanoribbon on the MoSe₂ substrate was adopted to simulate the CrSe₂/MoSe₂ vertical heterostructure. As shown in Fig. R6b, the DFT-calculated local density of states (LDOS) plot of the MoSe₂ surface exhibits that the upward band bending is not localized at vicinity of edges but exists at the whole MoSe₂ area under the CrSe₂ nanoribbon. The effect of edge states can also be ruled out by the previously reported work (*Nat. Commun.* 7, 10349, 2016) in which there is little band bending in the monolayer WSe₂ or MoSe₂ region despite the existence of edge states at the bilayer-monolayer WSe₂ or MoSe₂ interfaces. Therefore, the band bending in the first-layer MoSe₂ is mainly due to the interlayer charge transfer.

Fig. R6 | DFT-calculated LDOS plot of the MoSe₂ surface in the CrSe₂/MoSe₂ vertical heterostructure.

Change12. Added two panels (Fig. R6a and 6b) in Supplementary Fig. 8 exhibiting the DFT-calculated band bending in the first-layer MoSe₂.

Change13 (Page 8). Added the sentence “The upward band bending in the first-layer MoSe₂ due to the interlayer charge transfer can also be reproduced by DFT calculations (Supplementary Fig. 8)” at the end of paragraph .

Reviewer #3 (Remarks to the Author):

This study experimentally demonstrated in-plane MoSe₂ template induced selective growth of H-phase semiconducting CrSe₂ and the decisive role of in-plane heteroepitaxial template with interfacial energy by lattice matching. The visualization of band profiles of the in-plane heterostructures between MoSe₂ and CrSe₂ combined with theoretical calculations seem attractive because a universal integration scheme for 2D/2D lateral heterostructures can guarantee the possibilities of design and fabrication of various (opto-)electronic 2D devices. In general, this is a rigorous and well-organized work and the findings are of considerable interest to scientists working in the 2D materials/devices field. However, this reviewer noticed that the concept of in-plane template induced selective growth has already been demonstrated in previous studies and well developed. Thus, this reviewer suggests the authors to convincingly illustrate the novelty of the current work. At least, the following issues should be careful addressed:

Reply: We thank the review for the positive evaluation and constructive suggestions. The template-synthesis method has indeed been extensively used for growing different kinds of nanomaterials, including metal, metal oxides, metal-organic frameworks (MOFs). As for 2D TMDs, many lateral TMD heterostructures (*e.g.*, WS₂-MoS₂, WS₂-WSe₂, WSe₂-MoS₂) have also been synthesized through a CVD method. However, to our knowledge, it has been rarely studied by utilizing the in-plane templates to induce phase-selective growth of 2D-TMDs. We have also achieved the MBE growth of H-phase VSe₂ through utilizing the in-plane template of MoSe₂ nanoribbons. We believe this method can be a promising and universal approach for phase engineering of 2D TMDs, thereby further expanding the library of crystal phases and promoting the property research of specific phases.

i) First of all, it is better to provide the underlying mechanism related to basic nucleation and growth mechanism of 1D MoSe₂ nanoribbons at the reported growth conditions (*i.e.*, Why can the MoSe₂ crystals be successfully obtained in the form of 1D nanoribbons?)

Reply: The growth temperature has been demonstrated to be one of the key factors in the MBE growth of MoSe₂ nanoribbons by previous works (*Nat. Commun.* 8, 15135, 2017; *Adv. Mater.* 29, 1605641, 2017). When the substrate temperature is lower than 250 °C, the MBE-grown MoSe₂ flakes have fractal shapes (as shown in Fig. R7a) due to the lower mobility of adatoms around the island edges. As the temperature is elevated, 2D MoSe₂ islands with higher crystallinity can be obtained (Fig. R7b). Meanwhile, the density and length of MTBs in the MoSe₂ monolayers can be controlled by tuning the growth temperature. More isolated and longer MTBs can be obtained at higher temperature. The MoSe₂ nanoribbons with well-defined orientations form at the substrate temperature kept at about 550 °C. Apart from the growth temperature, the Se:Mo flux also has the equivalent effects on the growth favouring nanoribbon growth at lower Se concentrations. The evolution of MoSe₂ island shapes is determined by the relative energies and growth rates of the different edge structures. The atomic growth mechanism of MoSe₂ nanoribbon has been revealed by DFT calculations in the reported

work (*Nat. Commun.* 8, 15135, 2017). The armchair edges were calculated to have the higher energy than the zigzag (Mo- and Se- terminated) edges. The armchair edges grow much faster than the zigzag edges, which results in the ribbon structures.

Fig. R7 | Temperature-dependent morphology of MBE-grown MoSe₂. **a,b**, STM image of the MoSe₂ monolayer grown at about 250 °C ($V_s = 2$ V, $I_t = 20$ pA) and 400 °C ($V_s = -2.4$ V, $I_t = 10$ pA). **c**, The corresponding atomic-resolution STM image of MTBs in **a** ($V_s = -0.8$ V, $I_t = 300$ pA). **d**, STM image of MTBs in the MoSe₂ monolayer ($V_s = -0.8$ V, $I_t = 500$ pA).

Change14. Added three panels (Fig. R7a, 7b and 7c) in Supplementary Fig. 1 exhibiting the temperature-dependent morphology of MBE-grown MoSe₂.

Change15 (SI Page 1). Added the sentence “The growth temperature has been demonstrated to be one of the key factors in the MBE growth of MoSe₂ nanoribbons by previous works^{1,2}. When the substrate temperature is lower than 250 °C, the MBE-grown MoSe₂ flakes have fractal shapes (as shown in Fig. S1a) due to the lower mobility of adatoms around the island edges. As the temperature is elevated, 2D MoSe₂ islands with higher crystallinity can be obtained (Fig. S1b). Meanwhile, the density and length of MTBs in the MoSe₂ monolayers can be controlled by tuning the growth temperature. More isolated and longer MTBs can be obtained at high temperature (Fig. S1d). The MoSe₂ nanoribbons with well-defined orientations form at the substrate temperature kept at about 550 °C (Fig. S1e). Apart from the growth temperature, the Se:Mo flux also has the equivalent effects on the growth favouring nanoribbon growth at lower Se concentrations. The evolution of MoSe₂ island shapes is determined by the relative energies and growth rates of the different edge structures. The atomic growth mechanism of MoSe₂ nanoribbon has been revealed by DFT calculations in the reported work¹. The armchair edges were calculated to have the higher energy than the zigzag (Mo- and Se- terminated) edges. Therefore, the armchair edges grow much faster than the zigzag edges, which results in the ribbon structures.” in the **Supplementary Note 1**.

ii) The authors are also asked to provide the detailed information on the structural properties (width, thickness, etc.) of the 1D MoSe₂ templates depending on the growth parameters (at their optimized growth regime) owing to their great important role in the lateral heteroepitaxy and please further explain what the main variables were that were able to control the thickness of the single-crystalline 1D MoSe₂ at the monolayer level.

Reply: The width control of MBE-grown MoSe₂ nanoribbons through growth temperature has been demonstrated by previous works (*Adv. Mater.* 29, 1605641, 2017), in which the width will gradually decrease as the growth temperature is elevated. The same tendency in the growth of MoSe₂ nanoribbons can also be observed in our experiments. The thickness can be controlled by tuning the growth duration. The density control of MoSe₂ nanoribbons can be achieved by tuning the growth parameters of flux rate and growth duration. At the optimized growth parameters of growth temperature and Se:Mo flux, the 1D MoSe₂ nanoribbons are able to be controlled at the monolayer level by tuning the flux rate and growth duration. Regarding to MoSe₂ nanoribbons grown at 550 °C in our experiments, the statistical analysis of width and layer numbers was carried out (Fig. R8). The widths of MoSe₂ nanoribbons are mostly distributed from 10 to 25 nm. The statistical analysis on thickness distribution indicates the MoSe₂ nanoribbons are mostly monolayer and bilayer.

Fig. R8 | Statistical analysis of ribbon width and thickness of MoSe₂ grown at 550 °C.

Change16. Added two panels (Fig. R8a and 8b) in Supplementary Fig. 1 exhibiting the statistical analysis of ribbon width and thickness of MoSe₂ grown at 550 °C.

Change17 (SI Page 2). Added the sentence “The width control of MBE-grown MoSe₂ nanoribbons through growth temperature has been demonstrated by previous works, in which the width will gradually decrease as the growth temperature is elevated². The same tendency in the growth of MoSe₂ nanoribbons can also be observed in our experiments. The thickness can be controlled by tuning the growth duration. The density control of MoSe₂ nanoribbons can be achieved by tuning the growth parameters of flux rate and growth duration. At the optimized growth parameters of growth temperature and Se:Mo flux, the 1D MoSe₂ nanoribbons are able to be controlled at the monolayer level by tuning the flux rate and growth duration. Regarding to MoSe₂

nanoribbons grown at 550 °C in our experiments, the statistical analysis of width and layer numbers was carried out (Fig. S1g and 1h). The widths of MoSe₂ nanoribbons are mostly distributed from 10 to 25 nm. The statistical analysis on thickness distribution indicates the MoSe₂ nanoribbons are mostly monolayer and bilayer.” in the Supplementary Note 1.

iii) In Fig. 1f, why are the edges of as-grown H-CrSe₂ crystals strongly jagged and not atomically straight during MBE growth? (and what was the lateral growth rate of H-CrSe₂?) In addition, why does CrSe₂ prefer edge-mediated growth compared to surface-mediated growth? Not even the presence of CrSe₂ island above the MoSe₂? The reasons must be explained in detail.

Reply: The H-phase CrSe₂ was grown at the substrate temperature of about 200 °C which is much lower than the growth temperature of 550 °C for fabricating MoSe₂ nanoribbons. The lower mobility of adatoms around the island edges might result in the formation of jagged edge structures. Besides the jagged edge structures, the small-scale straight edge structures can also be observed in the same sample. The lateral growth rate of H-phase CrSe₂ is about 0.02 ML/min in our experiments. The edges of MoSe₂ nanoribbons can act as the active sites for the epitaxial growth due to the existence of dangling bonds. Compared to surface-mediated growth, the edge-mediated growth is more thermodynamically favorable. As for the growth of TMD lateral heterostructures, there are also DFT calculations revealing the adsorption energies of radicals at the edges are much lower than that on the surface (*ACS Nano* 13, 885–893, 2019). Therefore, the radicals prefer to adsorb at the edges to form the seed points rather on the surface. When we increase the deposition duration of CrSe₂, the isolated CrSe₂ islands with the non-layered structures grown on the top of MoSe₂ can also be observed in our experiments (as shown in Fig. R9c and 9d).

Fig. R9 a,b, The MoSe₂-CrSe₂ lateral heterostructures with small-scale straight edge structures. **c,d**, STM images of isolated 1T-CrSe₂ islands grown on the top of MoSe₂ ($V_S = -1.8$ V, $I_t = 30$ pA).

Change18 (page 4). Added “due to the existence of dangling bonds at the edges” into the sentence “The diffusing atoms preferentially nucleate and aggregate at the active edges of MoSe₂ nanoribbons.” in the paragraph (page 4).

Change19 (page 4). Added the sentence “The lateral growth rate of H-phase CrSe₂ is about 0.02 ML/min in our experiments.” in the **Methods**.

Change5. Added two panels (Fig. R9c and 9d) in Supplementary Fig. 2 exhibiting the samples of isolated CrSe₂ islands with the non-layered structures grown on the top of MoSe₂.

iv) Page 7, line 189: Why does CrSe₂/MoSe₂ vertical heterostructure form and under what growth conditions? In this case, I believe the second-layer CrSe₂ is H-phase based on the supporting STS study and then how H-CrSe₂ crystals can be grown on MoSe₂ instead of the growth T-CrSe₂? And the authors claim it follows a layer-by-layer growth mode as compared to the lateral growth of H-CrSe₂. Though, no details on growth conditions and underlying mechanism are provided.

Reply: The formation of CrSe₂/MoSe₂ vertical heterostructures is based on the bilayer MoSe₂ nanoribbons. The second-layer (SL) MoSe₂ nanoribbons usually have the narrower width than the first-layer (FL) nanoribbons. The epitaxial growth of H-phase CrSe₂ will take place simultaneously on the both layers. The H-phase SL-CrSe₂ can be observed to seamlessly connect to the in-between SL-MoSe₂ nanoribbons (as shown in Fig. R10). The representative CrSe₂/MoSe₂ vertical heterostructures are formed with the second layer being the H-phase CrSe₂ and first layer being the MoSe₂ in the bilayer structures.

Fig. R10 | Large-scale STM image of the CrSe₂/MoSe₂ vertical heterostructure formed with the second-layer CrSe₂ and first-layer MoSe₂ in the vertical structures.

Change20 (page 7). Replaced the sentence “Owing to layer-by-layer growth mode, the CrSe₂/MoSe₂ heterostructures are formed with the first-layer MoSe₂ and second-layer CrSe₂ in the bilayer structures. This is representative of a CrSe₂/MoSe₂ vertical heterostructure.” to “The formation of CrSe₂/MoSe₂ vertical heterostructures is based on the bilayer MoSe₂ nanoribbons. In the bilayer MoSe₂ nanoribbons, the second-layer (SL) MoSe₂ nanoribbons usually have the narrower width than the first-layer (FL) nanoribbons. The epitaxial growth of H-phase CrSe₂ will take place simultaneously on the both layers with the SL-CrSe₂ seamlessly connected to the in-between SL-MoSe₂ nanoribbons (Fig. 3e). The representative CrSe₂/MoSe₂ vertical heterostructures are formed with the second layer being the H-phase CrSe₂ and first layer being the MoSe₂ in the bilayer structures.” in the paragraph.

Change21 (Fig. 3e). Increased the image contrast to clearly exhibit the SL-CrSe₂ and SL-MoSe₂ nanoribbons. The FL-CrSe₂, FL-MoSe₂, SL-CrSe₂ and SL-MoSe₂ were labelled in Fig. 3e.

i) The quantum-confined TLL behavior depending on the length of MTBs, quasi-1D metals, in H-phase CrSe₂ needs to be explained in detail, which will provide direction toward further research and device applications.

Reply: The quantum-confined TLL behavior depending on the length of MTBs was discussed in the **Supplementary Note 4**. For 2D or 3D metallic systems, the electronic behavior can be described by Landau Fermi liquid (FL) theory of non-interacting quasiparticles. When the electrons are confined in 1D systems, the quasiparticle excitation mechanism breaks down and electrons become a strongly correlated quantum liquid obeying the Tomonaga-Luttinger liquid (TLL) behavior^{6,7}. As (quasi) one-dimensional metallic systems, TLL behavior has been revealed in the MTBs. As shown in Fig. R11, the gap size increases with the length of MTBs getting shorter, which is the signature of TLL behavior. In the TLL theory, the energy gap of the finite system with length L can be described as $E_{\text{gap}} = [(\pi v_c/2K_c) + (\pi v_s/2K_s)](1/L)$, where v_c and v_s stand for the velocity of charge and spin excitation, respectively. Two Luttinger parameters K_c and K_s encode the interaction strength. Another signature of TLL behavior is the spin-charge separation which has the distinct dispersions of spin and charge excitations with velocities v_s and v_c . In the Fourier transformation of 2D plot of dI/dV spectra which can directly reveal the dispersion of confined states, two linear dispersion branches with different slopes corresponding to the spin and charge density excitations can be observed.

Fig. R10 | Electronic properties of MTBs in the H-phase CrSe₂ monolayer. **a**, High-resolution STM image of the MTB with a length of ~12.8 nm ($V_s = 0.2$ V, $I_t = 10$ pA). **b**, The dI/dV spectra taken on the MTB with different length.

Change22 (page 9). Added the sentence “In the TLL theory, the energy gap of the finite system with length L can be described as $E_{\text{gap}} = [(\pi v_c/2K_c) + (\pi v_s/2K_s)](1/L)$, where v_c and v_s stand for the velocity of charge and spin excitation, respectively. Two Luttinger parameters K_c and K_s encode the interaction strength.

The STS spectra acquired on the MTB exhibit the metallic character with low-energy states around the Fermi level (Fig. 4e).” in the paragraph.

REVIEWERS' COMMENTS

Reviewer #1 (Remarks to the Author):

The authors have done an outstanding job answering my questions and comments in the previous review. I do not have further questions and I support this paper to be published in Nature Communications.

Reviewer #2 (Remarks to the Author):

The authors have addressed the question properly and I recommend its publication now

Reviewer #3 (Remarks to the Author):

I reviewed the revised paper and letter and this revision has addressed most of my concerns. No additional review is necessary and this paper is now publishable in Nature Communications.